# High-performance chemical- and light-inducible recombinases in mammalian cells and mice

Benjamin H. Weinberg [1], Jang Hwan Cho[1], Yash Agarwal[1], N.T.Hang Pham[1], Leidy D. Caraballo[1], Maciej Walkosz[1], Charina Ortega [1], Micaela Trexler [1], Nathan Tague[1], Billy Law[1], William K.J. Benman[1], Justin Letendre[1], Jacob Beal[2]* & Wilson W. Wong [1]*

Site-specific DNA recombinases are important genome engineering tools. Chemical- and light-inducible recombinases, in particular, enable spatiotemporal control of gene expression. However, inducible recombinases are scarce due to the challenge of engineering high performance systems, thus constraining the sophistication of genetic circuits and animal models that can be created. Here we present a library of >20 orthogonal inducible split recombinases that can be activated by small molecules, light and temperature in mammalian cells and mice. Furthermore, we engineer inducible split Cre systems with better performance than existing systems. Using our orthogonal inducible recombinases, we create a genetic switchboard that can independently regulate the expression of 3 different cytokines in the same cell, a tripartite inducible Flp, and a 4-input AND gate. We quantitatively characterize the inducible recombinases for benchmarking their performances, including computation of distinguishability of outputs. This library expands capabilities for multiplexed mammalian gene expression control.

[1] Department of Biomedical Engineering and Biological Design Center, Boston University, Boston, MA 02215, USA. [2] Raytheon BBN Technologies, Cambridge, MA 02138, USA. *email: jakebeal@ieee.org; wilwong@bu.edu

Site-specific recombinases (SSRs) are enzymes that perform site-specific DNA recombination through their ability to carry out DNA cutting and ligation operations[1]. SSRs can effectively regulate gene expression based on the placement of their corresponding set of recombination sites around a gene of interest or transcription termination sequence. Furthermore, chemical- and light-inducible systems have been engineered to control the expression or activity of SSRs, enabling spatio-temporal control of gene expression[2–4]. Due to their unique chemistry and performance, SSRs have been used extensively in genetic circuit designs[5–9] and animal model development[10–12] to interrogate biological processes and design principles. While inducible recombinases have proven to be invaluable in many studies[13–15], especially in mammalian cells, the number of orthogonal recombinases available is limited. Most inducible systems are optimized solely for Cre recombinase and few, if any, orthogonal systems are available for other recombinases. This limitation constrains the sophistication of the genetics that can be developed and ultimately the biological and engineering problems that can be addressed. Furthermore, some small-molecule inducers (e.g., antibiotics or hormone analogs) may not be compatible with some experimental objectives. As such, there is an urgent need to develop a larger collection of inducible recombinases that employ an array of orthogonal inducers.

A major challenge in developing inducible recombinases in mammalian cells is to ensure both low basal activity and high induced activity, which requires orthogonal SSRs and induction systems to work effectively together. We and others have shown that many recombinases are active in mammalian cells[7,16], thus serving as a potential source for engineering orthogonal-inducible recombinases. Furthermore, a variety of inducible systems has been used to control the activity or expression of Cre. While the availability of diverse classes of SSRs and inducible systems provide a rich pool of source materials, engineering optimal pairings of SSRs and inducible systems is challenging and requires systematic testing and characterization. One proven and attractive approach to generating inducible recombinases has been to utilize chemical-inducible dimerization (CID) domains.

CID domains, whereby a protein of interest is split into two fragments and each one fused to a complementary chemical-inducible heterodimerization domain, are an attractive system to control the activity of recombinases. The most prominent CID system is the FKBP/FRB (FK506 binding protein/FKBP rapamycin binding) system, which forms a heterodimer upon the addition of the rapamycin analog, AP21967, termed rapalog (RAP). A study utilized the FKBP/FRB system to generate a rapamycin-inducible Cre recombinase[17]. Since this work, orthogonal CID systems have been identified, including one that uses phytohormone abscisic acid[18] (ABA) and another that uses phytohormone gibberellin (GIB)[19,20]. CID systems are advantageous because split proteins tagged with CIDs can be rendered non-functional until the molecule is administered, having the potential to yield low basal activities. Thus, high levels of the split protein can be expressed in advance in the desired cellular compartment, enabling fast responses. This is in contrast to other post-translational systems that are either actively degrading the protein until a drug is administered to stabilize the protein[21] or sequestering the protein to the cytoplasm until drug exposure permits translocation of the protein to the nucleus[22,23]. Furthermore, the expression of CID-regulated recombinases can be controlled by conditional promoters (e.g., tissue-specific promoters) independent of the CID system. This feature enhances spatiotemporal control of gene expression[24], which is not achievable if recombinase expression is controlled by a drug-inducible promoter (e.g., Tet-ON/doxycycline[25]).

In this work, we address the shortage of inducible recombinases by developing a suite of inducible recombinases through screening for functional split locations to use with the (RAP), ABA, and GIB CID systems. In total, we are able to achieve functional chemical induction in six recombinases. Moreover, we demonstrate the extensibility of this approach by engineering two light- and temperature-inducible recombinases using light-inducible dimerization (LID) domains for conditional recombination. Finally, we demonstrate the functionality of these engineered systems by implementing the first 4-input drug-inducible logic gates and tripartite-inducible Flp recombinase in mammalian cells. To ensure accessibility of the inducible recombinases we have developed, we have deposited our library to non-profit plasmid repository Addgene. We perform detailed characterization of select recombinases using community-derived standards and metrics for benchmarking purposes and quantitative comparison. Together, our library of characterized inducible recombinases will enable the creation of genetic circuits and animal models for both fundamental studies and biotechnological applications.

## Results

**Selecting split locations**. CID systems were utilized for small-molecule control of recombinases. Three recombinases from the tyrosine recombinase family were chosen, including two from phage (Cre and VCre), one from yeast (Flp), and three from the large serine integrase family (φC31, TP901, and Bxb1). These recombinases were chosen due to their high activity in mammalian cells and orthogonality with one another[7,16]. Primary and secondary sequences were first identified in order to identify potential split sites. A crystal structure was available for Cre (Protein Database (PDB): 1XO0), which was utilized to determine secondary structure of analogous VCre through HHPred alignment (Supplementary Fig. 1). Secondary structure annotations for serine integrases were determined from a literature source[26] that utilized TP901 integrase N-terminus domain structure and A118 integrase C-terminus domain structure. In general, split sites were chosen to avoid secondary structures and catalytic residues, as we hypothesized that introducing CID domains into these locations could permanently disrupt these structures and their roles in recombination (DNA binding, cleavage, exchange and ligation); however, many splits sites within these structures in Cre recombinase were chosen to counter-test this hypothesis. After a split location ($S$) was chosen, a starting fragment was cloned from the N-terminus of the protein to $S$ and a terminating fragment from $S + 1$ to the C-terminus of the protein. Fragments were inserted into vectors containing sequences to code for flexible glycine- and serine-rich linkers, nuclear localization signals, and CID domains (Supplementary Fig. 2). For the RAP system, FRB and FKBP CID domains were used; for the ABA system, PYL and ABI; and for the GIB system, GID1 and GAI. All secondary structure annotations, plasmid numbers, and fold changes are provided in the Source Data file.

**Screening for small-molecule inducible dimerization**. Screens for identifying functional split locations in tyrosine recombinases were performed using the GIB CID system (Fig. 1). In total, 42 locations were tested for Cre recombinase, 20 for Flp and 20 for VCre. The RAP CID system was used for 13 splits of φC31 and 11 for TP901, while the GIB CID system was used for screening of two Bxb1 split locations.

A variety of qualitative responses were found: functionally dead, yielding no activity with or without the addition of the drug; constitutively active, yielding partial or full recombination with no significant dependency on the drug; and successful responders,

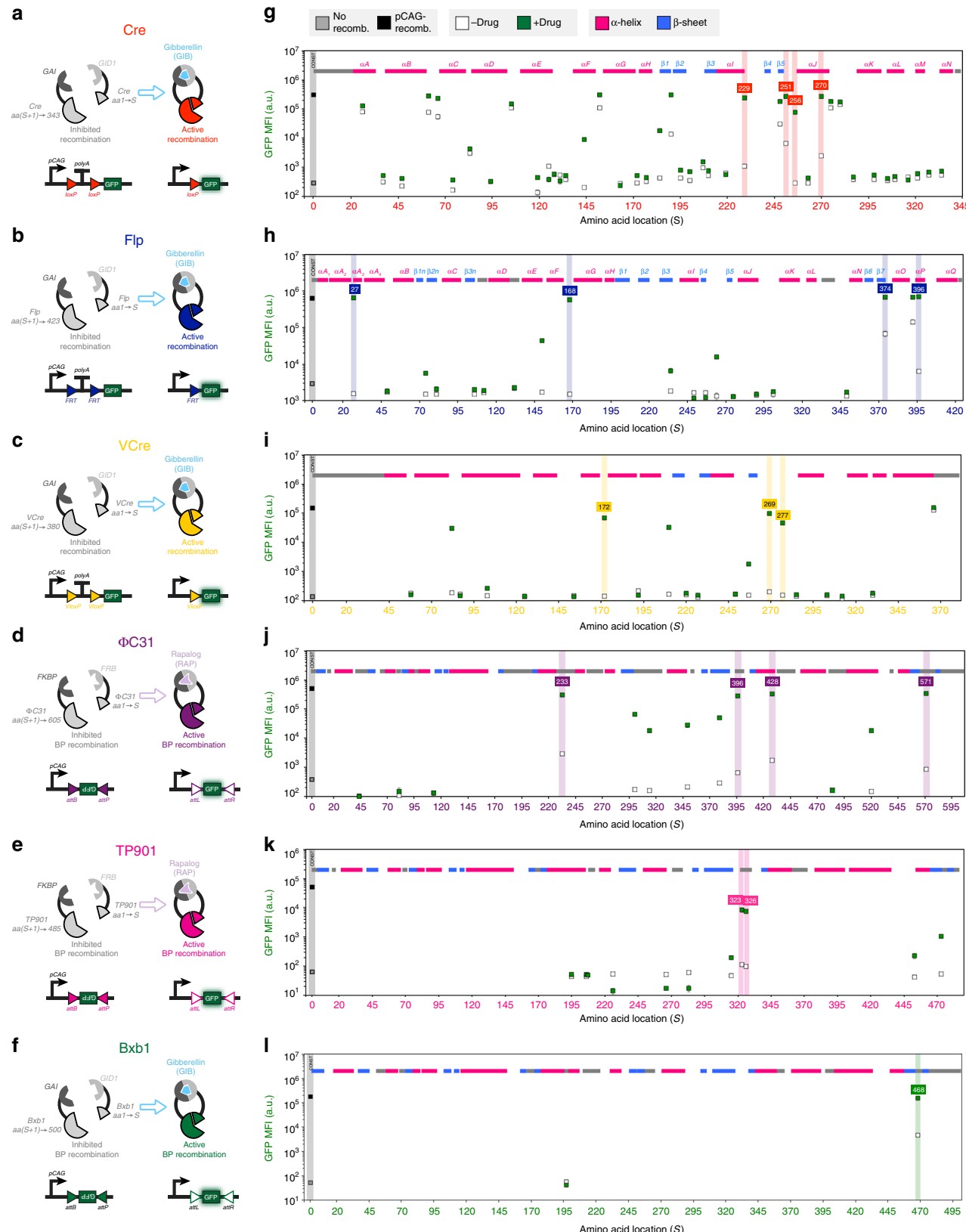

which saw a mild to strong dependency on the drug. A majority of the splits fall into the functionally dead category and these are thought to yield no activity due to improper formation of structures capable of effective recombination. In these cases, CID domains may have been inserted into places critical for recombination processes, or addition of CID domains caused too much three-dimensional contortion to perform recombination. Some locations yielded constitutively active responses, where CID domains apparently played little or no role in inhibiting and activating recombination. These configurations may splice off a region of the recombinase that may not be critical for a full recombination response. Moreover, resultant split halves

**Fig. 1** Screens of split locations yield small-molecule-dependent recombination responses. Cre, Flp, VCre, φC31, and TP901 recombinases (**a**–**f**) were split into two fragments at amino acid S and fused to gibberellin-inducible (GAI/GID1) or rapalog-inducible (FKBP/FRB) heterodimerization domains. Splits were made at various amino acid locations; pink, blue, and gray shaded regions indicate α-helices, β-sheets, or undetermined structures, respectively. Split recombinases were transfected along with reporters that yield green fluorescence protein (GFP) expression upon site-specific recombination (**g**–**l**). Constitutive recombinases (black square) or a blank vector (gray squares) were transfected to indicate highest or lowest expected GFP expression. Split recombinases were also transfected, and drug was either added (green squares) or not added (white squares) to the cell culture medium after 2 h. MFI indicates mean fluorescence intensity measured with arbitrary units (a.u.). Errors bars represent arithmetic standard error of the mean of three transfected cell cultures. Source data are provided as a Source Data file

of the recombinase could have enough self-association or recruitment through DNA binding to a recombination site to elicit a drug-independent response as well. Finally, some were successful responders, which triggered drug-dependent responses. Within this group, a select few good responders yielded high fold activations with high green fluorescence protein (GFP) in the on state and low GFP in the off state. However, some less desirable poor/moderate responders also produced low levels of GFP expression in the on state or high levels of GFP in the off state.

Four split Flps with high responses to GIB ($S = 27$, 168, 374, 396) were studied in more detail through transfection of all combinations of each split component. All three components (N-terminal split half, C-terminal split half, and drug) were found to be necessary for recombination (Supplementary Fig. 3). Testing these split configurations without fusion to dimerization domains yielded no appreciable recombination except for one split location ($S = 374$), which was found to have high recombination activity in contrast to its low recombination response when fused with GIB-responsive GAI and GID1 CID domains.

Despite each recombinase having analogous structures, with Cre identifying most closely to VCre, successful split locations did not seem to show significant correspondence between recombinases. Interestingly, for Flp, φC31, and Bxb1, some locations that yielded strong drug-dependent responses were surprisingly close (<35 amino acids) to either the N-terminus or C-terminus. This result highlights the importance of the native protein structures in these terminal areas for recombination. Of the seven Cre splits ($S = 190$, 251, 184, 144, 270, 229, and 256) that yielded fold changes over ten, two ($S = 144$ and 270) were within secondary structures. Overall, 2 out of 19 (10.52%) of the Cre splits within secondary structures yielded fold changes over ten, and 5 out of 23 (21.74%) Cre splits outside secondary structures yielded fold changes over ten. Four of the best-performing Cre GIB candidates were compared against a split location determined from a previous work[17] where amino acids 19–59 were used for the starting fragment and 60–343 for the ending fragment. Despite placing GID1 and GAI domains at varying N-terminal or C-terminal ends for the 19–59/60–343 split, none were close to reaching the low off and high on responses of the newly identified splits (Supplementary Fig. 4).

**Domain swapping and benchmarking performance**. To gain better insight into the functionality of some of the inducible recombinases, quantitative flow cytometry analyses were performed to characterize performance. Fluorescence channel measurements were converted to units of molecules of equivalent fluorescein (MEFL)[27,28], a community-derived standard used to normalize measurements across instruments and fluorescent molecules (see Methods section for details). Analyses focused around splits 270, 27, and 233 of Cre, Flp, and φC31, respectively, with all three CID systems. In addition, the comparison also included a commonly used system in which Cre is fused to a mutated estrogen receptor (ER^T2)[2]; this complex remains in the cytoplasm until the addition of 4-hydroxytamoxifen (4OHT),

which induces translocation to the nucleus and results in SSR. A signal-to-noise (SNR) metric was used to evaluate how well the plus and minus drug conditions can be distinguished from each other; better separation between signals involves both a higher absolute difference in mean fluorescence intensities and also a lower spread of fluorescence values (noise) within those cell population distributions (Fig. 2a). Among the three recombinase systems, φC31 integrase yielded the highest SNR values; this could be attributed to the quieter reporter that is used, which necessitates the complete inversion of the GFP sequence; this is in contrast to using a polyA deletion strategy in the case of tyrosine recombinases, which yields a low but detectable amount of read-through for highly transfected cells (Fig. 2b, Supplementary Figs. 5–9). In the case of Cre and Flp RAP systems, however, SNR reached a maximum and then declined due to leaky recombination and noisier GFP expression in the off state. Over a time course of 100 h, leaky off behavior was especially apparent in Cre RAP, Cre 4OHT, and Flp RAP systems, which yielded high SNR values early on, but began to decline after 21 h. The GIB system consistently resulted in strong performance over all three recombinases and φC31 integrase performed well with all three CID systems. With respect to plasmid dosage, as correlated by transfection marker expression level, fold activation (mean on divided by mean off expression) and SNR values increased as on-state GFP fluorescence grew due to increased copies of reporter, while off-state GFP fluorescence remained low. However, RAP and 4OHT systems had noticeable values of transfection marker expression where fold activation and SNR peaked and then declined. This behavior is attributed to the significant rise in leaky off-state GFP expression with higher plasmid copy. Moreover, the 4OHT-inducible Cre performance had notable time dependence as well; even at mid-levels of transfection marker ($1 \times 10^7$–$4 \times 10^7$ MEFL), this system yielded significant decreasing performance over time. These observations highlight the importance of characterized operation ranges with respect to the expression level of biological device components and dynamic behavior over time.

An advantage of identifying viable split locations is that they can be applied to other into dimerization systems. Four splits for Cre, four for Flp, three for VCre, and four for φC31 having moderate to good responses from the initial screens were selected to swap for the other two CID domains, RAP and ABA (Fig. 2c–n, Supplementary Figs. 10–14). In general, swapping dimerization domains between GIB/RAP/ABA domains or fusion to different protein termini for the case of Flp recombinase (Supplementary Fig. 15) yielded drug-dependent responses for the selected split sites; however, many tyrosine recombinase split locations using the RAP system yielded surprisingly high levels of recombination without the addition of the drug, converting good responders with the GIB system into near constitutively active systems with RAP, particularly for higher values of transfection marker expression.

**LID and temperature-inducible dimerization**. A variety of LID systems are available for applications that necessitate minimal perturbation and precise stimulation. Here, we chose to use a blue LID system, termed Magnet, which uses heterodimerization

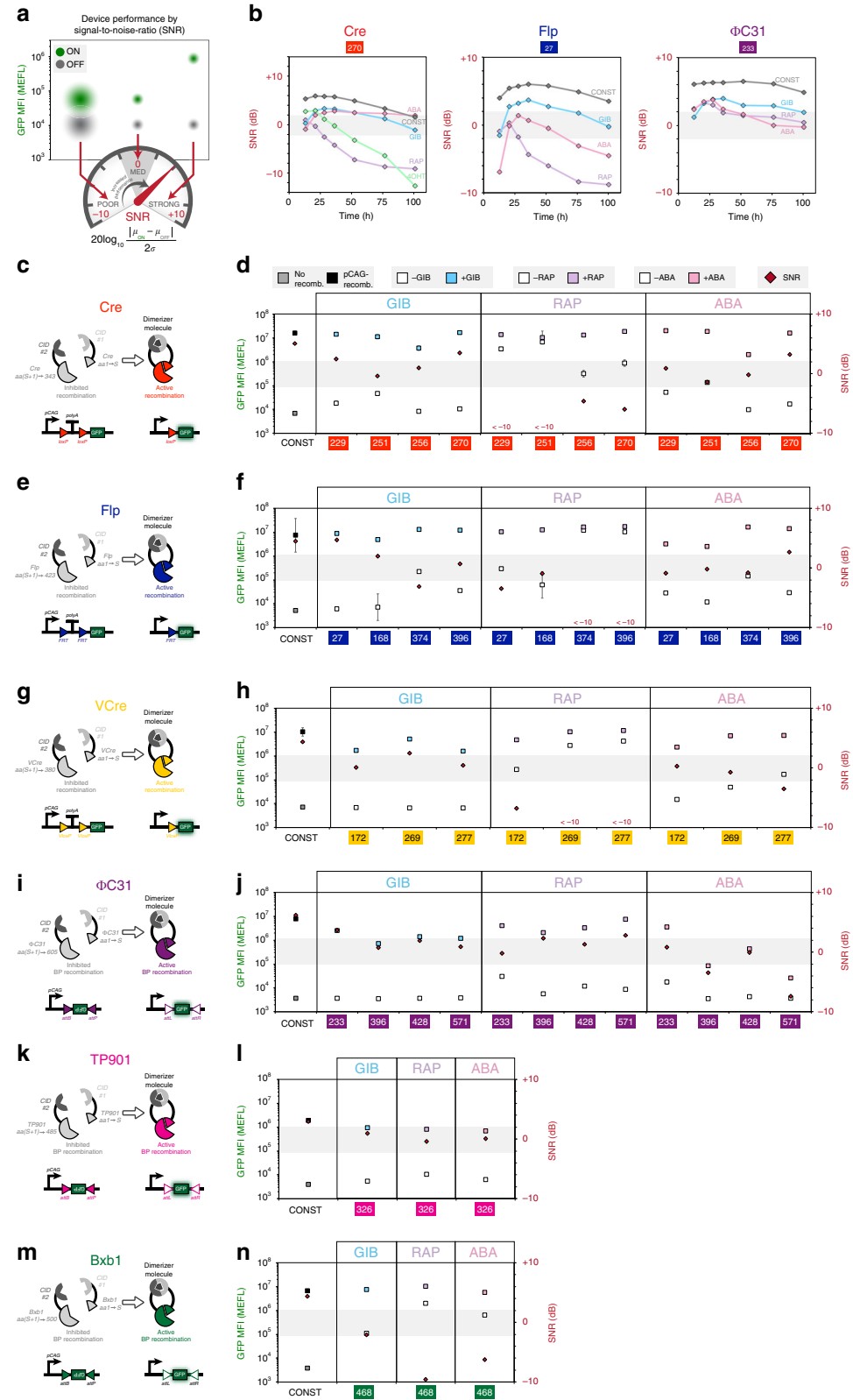

domains (pMag and nMag) engineered from the blue light photoreceptor VVD from the filamentous fungus *Neurospora crassa*. A programmable blue light-emitting device was used to induce four splits of Cre or Flp that use the pair of Magnet LIDs (Fig. 3). In addition, a construct that expresses an established[3] Cre Magnet (termed PA-Cre) using the 19–59/60–343 split was tested in parallel. Sixty-second blue light pulses were administered every

hour over a 24-h period, 18 h post transfection. Blue LID was achieved for all constructs for Cre and Flp, albeit with different performances amongst splits. Serendipitously, we found that exposing the cells to colder temperatures could also induce recombination, in some cases more than blue light. Temperature induction was performed by exposing the cells to either room temperature (22 °C) or refrigerated (4 °C) conditions for four

**Fig. 2** Swapping chemical-inducible dimerization domains yields a large repertoire of small-molecule inducible recombinases. **a** A signal-to-noise (SNR) metric can be used to capture distinguishability of on and off states, accounting for both the absolute difference in mean signal expression and spread (noise) of the distributions. SNR dynamics in units of decibels (dB) were captured over 100 h (**b**) post transfection for particular amino acid splits of inducible Cre, Flp, and φC31 systems incorporating the gibberellin (GIB), rapalog (RAP), and abscisic acid (ABA) dimerization domains GAI/GID1, FKBP/FRB, and PYL/ABI, respectively. A 4-hydroxytamoxifen (4OHT)-inducible pCAG-ER^{T2}-Cre-ER^{T2} construct is also included in **b**. Various splits of Cre (**c**, **d**), Flp (**e**, **f**), VCre (**g**, **h**), φC31 (**i**, **j**), TP901 (**k**, **l**), and Bxb1 (**m**, **n**) were incorporated with the GIB-, RAP-, and ABA-associated dimerization domains. Measurements of GFP mean fluorescence intensity (MFI) in units of molecules of equivalent fluorescein (MEFL) and SNR were made for plus drug (colored blue, purple, and pink squares corresponding to gibberellin, rapalog, and abscisic acid added 2 h post transfection) and minus drug (white squares) conditions. Constitutive recombinases (black squares) and blank vectors (gray squares) were also transfected as a comparison. Red diamonds indicate SNR score. Error bars of MFI indicate the geometric standard deviation of means for three transfected cell cultures. Source data are provided as a Source Data file

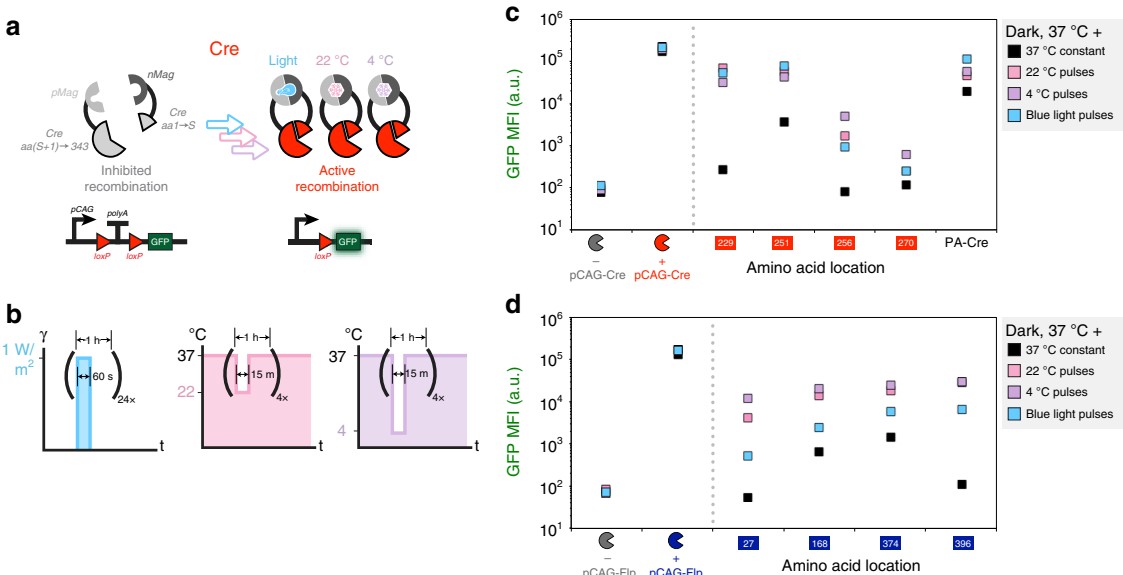

**Fig. 3** Light- and temperature-inducible recombinases. **a** A schematic showing a split Cre fused to nMag and pMag blue light-inducible dimerization domains. **b** Pulses indicating the changing light (left) and temperature (center and right) conditions. Plots indicate GFP mean fluorescence intensity (MFI) of various nMag/pMag-tagged split Cre (**c**) and Flp (**d**) recombinases under blue light (blue squares), room temperature (pink squares), and refrigeration (purple squares). Black squares indicate samples treated with constant temperature and dark conditions. PA-Cre indicates expression of a previously developed blue light-inducible Cre recombinase[3]. Inducible recombinases were compared with samples transfected with constitutive recombinases or a blank vector. Error bars of MFI indicate the arithmetic standard error of the mean between three transfected cell cultures. Source data are provided as a Source Data file

hourly 15-min pulses 18 h after transfection. The temperature sensitivity of this system may not be completely surprising as VVD has been found to play a role in temperature regulation[29,30] and others have engineered another closely related blue LID system to be more sensitive to colder temperature[31].

**Protein and DNA-based logic**. With this repertoire of inducible recombinases capable of responding to different drugs and having various functional split locations, we hypothesized that we could generate multi-input logic gates. The first approach taken was to use multiple CIDs within one recombinase protein. Flp recombinase was split into three segments using two functional split locations found previously ($S_1 = 27$, $S_2 = 396$); the GIB and ABA CID domains were utilized for small-molecule-mediated assembly of this tripartite double-split complex (Fig. 4a). A strong response was found only when both GIB and ABA were administered, demonstrating a 2-input AND gate functionality with 125-fold increase in expression upon application of both drugs (Fig. 4b); however, this did not work for another pair ($S_1 = 168$, $S_2 = 396$) of vetted split locations (Supplementary Fig. 16). A second approach taken to achieve multi-input logic operations was to use multiple inducible recombinase systems and integrate those signals with a multi-input

reporter DNA plasmid (Fig. 4c). Three inducible split-recombinase systems and a 4OHT-inducible Cre construct were selected. These signals were then integrated using a 4-input AND gate reporter where GFP is expressed only when two termination sequences are deleted and both promoter and GFP sequences are inverted. All combinations of drugs were applied to the cells and the highest GFP expression was found when all four drugs were applied together (>250-fold compared to no drugs) (Fig. 4d).

**Applications of split recombinases**. The inducible expression of genes using recombinases has wide applications. We demonstrate the multiplexed control of three genes that code for therapeutically relevant cytokines using a genetic switchboard consisting of three inducible recombinases and three reporter genes. In this system, we utilize high-performing ABA-inducible Cre, GIB-inducible Flp, and RAP-inducible φC31 systems we developed along with Cre-controlled interferon-γ (IFN-γ), Flp-controlled interleukin-10 (IL-10), and φC31-controlled IL-2 (Fig. 5a). IFN-γ and IL-2 are important inflammatory cytokines, especially against cancer. IL-10 is a pleotropic cytokine with both anti-cancer and anti-inflammatory potential. Supernatant cytokine measurement reveals strong activation of indicated cytokines

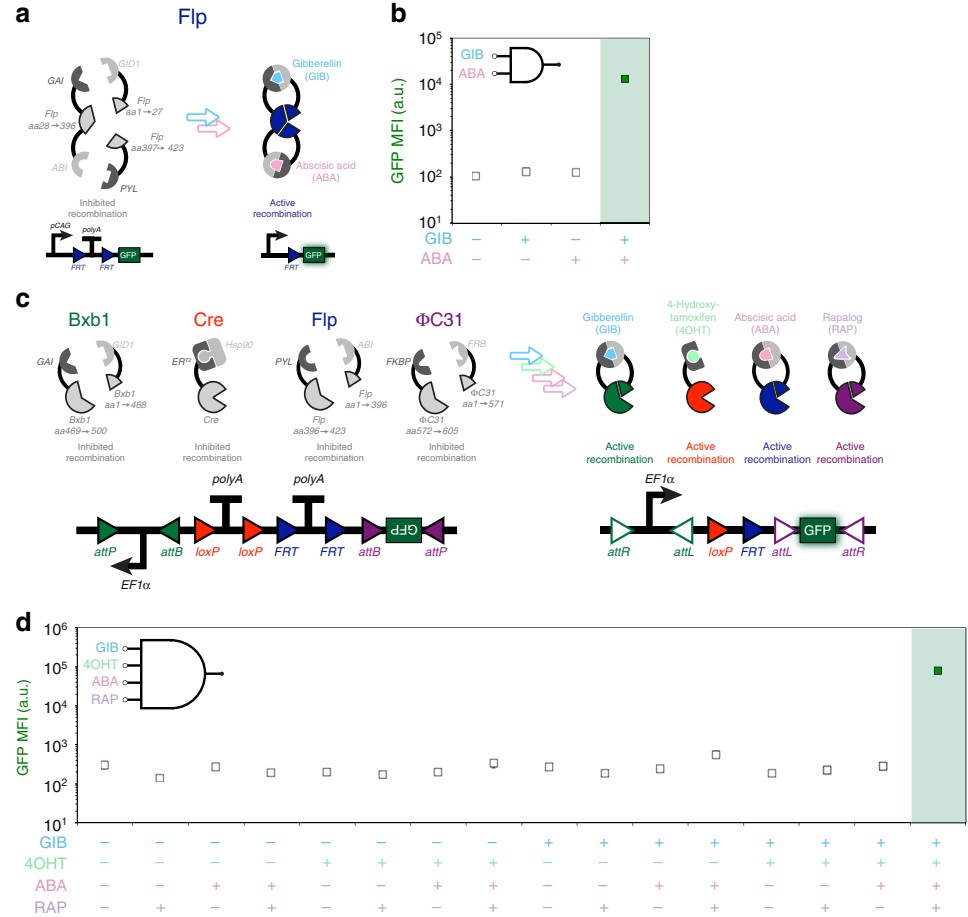

**Fig. 4** Protein and DNA-based recombinase logic gates. **a** A two-input protein-based AND gate is created by splitting Flp recombinase at two locations ($S_1 = 27$, $S_2 = 396$) and fusing to the gibberellin (GIB)- and abscisic acid (ABA)-associated chemical inducible dimerization domains. **b** Plotted results indicate GFP mean fluorescence intensity (MFI) of four conditions of GIB and ABA. **c** Four chemical-inducible recombinases can be used together to generate a DNA-based 4-input logic gate; these include a gibberellin-inducible split Bxb1 ($S = 468$), 4-hydroxytamoxifen (4OHT)-inducible ER[T2]-Cre-ER[T2], abscisic acid (ABA)-inducible split Flp ($S = 396$), and rapalog (RAP)-inducible split φC31 ($S = 571$). Drugs were added 2 h post transfection. **d** Plotted results indicate GFP. MFI of 16 conditions of GIB, 4OHT, ABA, and RAP. Error bars of MFI indicate the arithmetic standard error of the mean between three transfected cell cultures. Source data are provided as a Source Data file

with drugs added separately or in combination (Fig. 5b). In order to verify if our inducible recombinase systems could operate in the context of a living animal, we developed three xenograft luciferase mouse models (Fig. 5c). The models utilized ABA, GIB, or RAP-induced activity of Cre, Flp, and φC31, respectively, to activate the expression of firefly luciferase. Inducible recombinase systems and reporters were transfected into HEK293FT cells (Fig. 5d). The following day, transfected cells were transplanted subcutaneously into NSG mice. After another 24 h, drugs or vehicle-only controls were injected intraperitoneally. Finally, 24 h later, mice were given D-luciferin and imaged for luminescence. Imaging reveals elevation of luminescence for all three inducible recombinase systems (Fig. 5e) with ABA-Cre and RAP-φC31 systems yielding higher fold expression changes than the GIB-Flp system.

## Discussion

In this work, we have greatly increased the number of inducible recombinase systems available in mammalian cells. We demonstrate successful splitting of Cre, Flp, VCre, φC31, TP901, and Bxb1 recombinases and reconstitution of functionality with CID systems; this vastly enhances the potential usage of these recombinases, as few or zero inducible systems were available for each recombinase

previously. Additional split locations in Cre recombinase were found to yield higher performance than a previously recognized location; using the new split locations, we demonstrate GIB- and ABA-inducible Cre systems. Using viable split locations of Cre and Flp recombinases, we incorporate the blue light-inducible Magnet heterodimerization system. Moreover, we also found that we could use a reduction of temperature rather than light illumination to induce recombination with this system, providing yet another way of conditionally activating recombinase responses.

Not only did we find single functional split locations, we also found multiple viable candidates for most recombinases. Interestingly, each location yielded a different degree of performance depending on which CID system was utilized. For instance, Cre Magnet ($S = 270$) yielded surprisingly low sensitivity to blue light despite this split location having strong performance compared with other tested Cre splits using the CID domains. This finding highlights the importance of finding multiple functional split locations for each protein, as they may behave differently for different introduced dimerization domains. Moreover, the usage of multiple sites within one protein permits creation of multi-input logic gates. We demonstrate this idea with a doubly split Flp construct that requires both the addition of ABA and GIB to function. A tripartite complex such as this will be especially useful for intersectional targeting of cells. Each of the three fragments

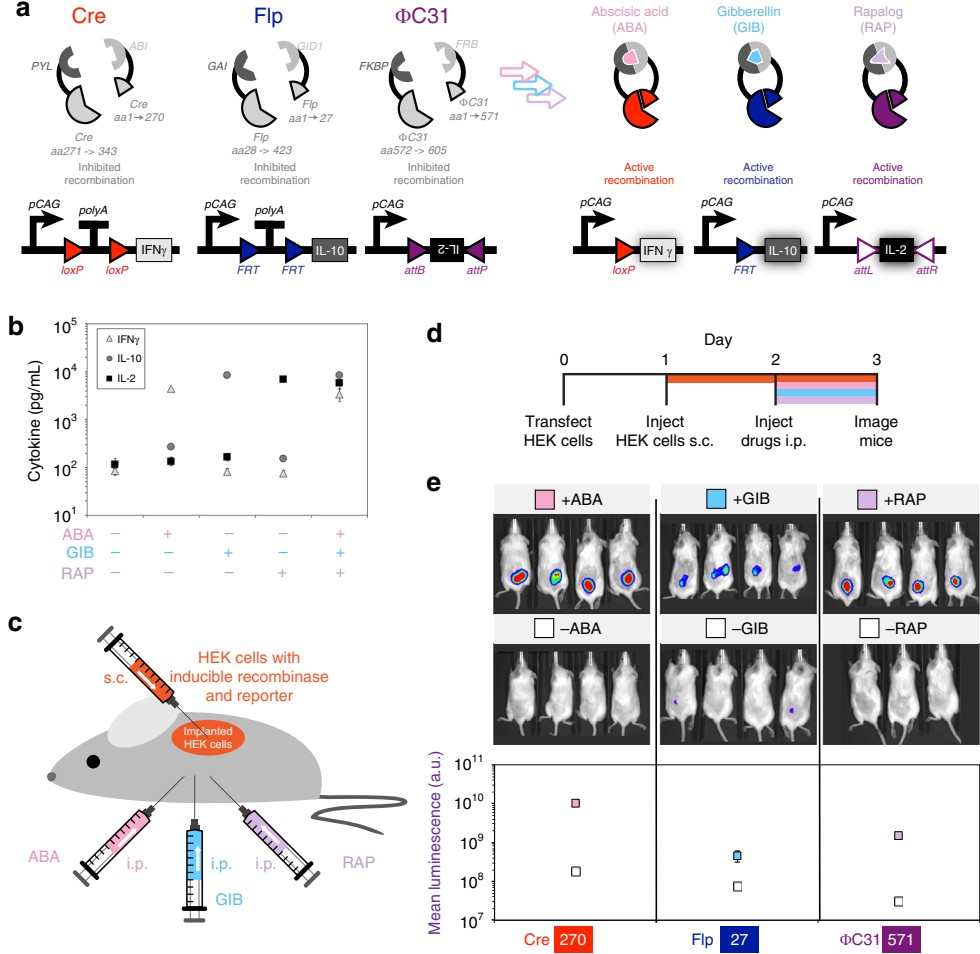

**Fig. 5** Applications of split recombinases. **a** A genetic switchboard using three inducible split recombinases to individually control expression of three therapeutically relevant human cytokines, interferon-γ (IFNγ), interleukin-2 (IL-2), and interleukin-10 (IL-10). Drugs were added 1 day post transfection. **b** Measurement of supernatant cytokine concentration using enzyme-linked immunosorbent assays ($n = 3$). **c** A xenograft mouse model of split recombinases used in **a** that were transfected in human embryonic kidney (HEK) cells alongside a corresponding luciferase reporter that expresses upon site-specific recombination. Transfected cells are transplanted subcutaneously (s.c.) and drugs or vehicle-only control are injected intraperitoneally (i.p.). **d** Time-course depicting time of transfection, injections, and imaging. **e** Images and averaged values of emitted luminescence from three split-recombinase systems. Error bars of mean cytokine and luminescence values indicate the arithmetic standard error of the mean. Source data are provided as a Source Data file

could be individually expressed by different promoters, thereby further tightening the desired spatial and temporal targeting of recombination. Moreover, we demonstrate a multi-output genetic switchboard that triggers control of three therapeutic proteins upon drug stimulation. Finally, we demonstrate that our inducible systems perform in live animal models, opening the doors to a wide range of applications.

In sum, we have created the largest-to-date library of well-curated inducible recombinases in mammalian cells for the genetic engineering community. Due to the high prevalence of SSR technology, we anticipate many applications and discoveries will come about using our inducible recombinase systems. As such, to ensure ease of access to our reagents, and encourage further development of these tools, we have deposited our reagents in non-profit plasmid repository Addgene. Newly discovered recombinases are being increasingly isolated[32], better high-throughput DNA-assembly techniques could provide faster and automated split site identification capabilities, and new ligand-responsive dimerization domains are waiting to be found or engineered.

## Methods
### Protein structural analysis and identification.
Secondary structure assignments for Cre were obtained from PDB crystal structure 1XO0_A. HHPred[33] was utilized

to predict secondary structures for VCre using Cre crystal structure data. ASA-View[34] was used to obtain absolute solvent area (solvent accessibility) from Cre. Secondary structure assignments for serine integrases φC31, TP901, and Bxb1 were adapted from a literature source[26]. 3D crystal structure images were created using the Visual Molecular Dynamics software package.

### DNA assembly.
A series of sub-cloning vectors containing CID or LID domains were assembled using standard molecular cloning techniques or Gibson isothermal assembly. For each CID or LID, two sub-cloning vectors were made such that a recombinase could be tagged with the domain on the C-terminus or N-terminus and a glycine-serine-rich linker L1 (SGGSGSGSSGGSGT). For insertion into a C-terminal CID or LID sub-cloning vector, recombinase starting fragments were first PCR amplified out of a full recombinase sequence template to contain a 5′ sequence to code for an *Mlu*I restriction site and a Kozak consensus sequence, and a 3′ sequence to code for a *Bsp*EI restriction site. These PCR fragments were then isolated using agarose gel extraction and purification (Epoch Life Science). Finally, the isolated fragments were digested with *Mlu*I and *Bsp*EI restriction enzymes (New England Biolabs), purified by PCR cleanup (Epoch Life Science) and ligated into a *Mlu*I/*Bsp*EI-digested CID and LID sub-cloning vectors that were isolated by PCR cleanup (Epoch Life Science). Reaction products were transformed into chemically competent Top10 *Escherichia coli* cells and selected on LB-agar plates with carbenicillin at 37 °C static conditions. Colonies were picked the following day, inoculated into carbenicillin-containing LB medium, and grown in a 37 °C shaking incubator (Infors). Once the cells were near or at stationary phase, plasmid DNA was isolated using mini plasmid preparation (Epoch Life Science). Analytical digests with *Mlu*I/*Bsp*EI were performed and run on gel electrophoresis to assay if a correct product was made. Plasmid clones with correct gel bands were sent for

sequencing (Quintara Biosciences). For insertion into a N-terminal CID or LID sub-cloning vector, recombinase terminating fragments were first PCR amplified out of a full recombinase sequence template to contain a 5′ sequence to code for an KpnI-HF restriction site and a 3′ sequence to code for an SV40 nuclear localization signal, stop codon, and an EcoRI-HF restriction site. A similar process was followed as before with the C-terminal CID and LID sub-cloning vectors; however, KpnI-HF and EcoRI-HF restriction digests (New England Biolabs) were carried out rather than with MluI and BspEI. To swap between CID and LID domains, sequenced fragments were gel isolated from sequenced vectors using MluI/BspEI or KpnI-HF/EcoRI-HF and inserted into corresponding CID or LID sub-cloning vectors. Occasionally, the BspEI site was blocked by dam methylation and PCR fragments were utilized instead. The extra central fragment for the trimeric Flp system was created with two different linkers L1 and L2 (GPAGGGSGGGSGGPAG) separating the Flp central fragment from the CID domains.

**Maintenance and transient DNA transfection of HEK293FT cells**. HEK293FT (Thermo Fisher) cells were kept in a static 37 °C humidified incubator with 5% $CO_2$, except for specified periods during temperature induction. Cells were maintained in Dulbecco's modified Eagle's medium medium (Corning) containing 5% fetal bovine serum (Thermo Fisher), 50 UI/ml penicillin and 50 μg/ml streptomycin mixture (Corning), 2 mM glutamine (Corning), and 1 mM sodium pyruvate (Lonza). A polyethylenimine (PEI) protocol was followed to transfect HEK293FT cells. A master stock of PEI was made by dissolving linear PEI (Polysciences, cat. no. 23966-2) in deionized water at a concentration of 0.323 g/L. Concentrated hydrochloric acid was used to drop the pH to 2–3 while stirring to ensure complete dissolution; sodium hydroxide was added over time to bring the pH to 7.5. The final solution was filter sterilized (0.22 μm), aliquoted, and stored at −80 °C. For transfections, HEK293FT cells were plated (250 μL) in tissue culture-treated 48-well plates (Fisher Scientific) 1 or 2 days prior such that they would be 50–70% confluent on the day of transfection. For most induction experiments in this work, 2000 ng (at 50 ng/μL) of DNA was prepared with specific combinations of plasmids and brought up to a volume of 100 μL using 0.15 M sodium chloride (NaCl, Fisher Scientific). Next, a PEI/NaCl mixture of 16 μL PEI stock and 84 μL NaCl was mixed with the DNA-NaCl solution and incubated at room temperature for at least 10 min. Finally, 25 μL of this solution was carefully added to each well of cells that were to be assayed; three wells for non-treated cells and three wells for drug or light treated. For time-course experiments, this was similarly followed, rather with eight times the amount of each component prior to dispensing of 25 μL into each well. DNA mass values for transfected plasmids are described in the Source Data file.

**Small-molecule chemical induction**. Stocks (1000X) of ABA (100 mM, Gold Biotechnology, cat. no. A-050-500) and GIB ester (10 mM, Toronto Research Chemicals, cat. no. G377500) were made in 100% ethanol and stored at −20 °C. A 1000X stock of RAP (A/C heterodimerizer AP21967) was obtained commercially (0.5 mM, Takara, cat. no. 635056) and stored at −20 °C. A 1000X stock of 4-OHT (1 mM, Sigma-Aldrich) was made in methanol and stored at −20 °C or −80 °C. These molecules were added to cell cultures such that the final concentration was 1× at the time of induction. For mouse experiments, stocks of ABA (40 mg/mL), GIB ester (4 mg/mL), and RAP (0.6 mg/mL, Takara, cat. no. 635055) were dissolved into vehicle solutions containing 16.7% propanediol, 22.5% PEG-400, and 1.25% Tween-80 through vigorous vortexing and water bath sonication. Vehicle components were purchased from Sigma-Aldrich (cat. nos. 82280, 91893, and P4780).

**Light illumination and temperature induction**. A modified large blue light transilluminator (IO Rodeo) was used in order to illuminate samples at 470 nm light. In order to control the timing and duration of light exposure, the included 18 V wall adapter was cut and exposed leads were used to power both an ARDUINO® UNO microcontroller and the light array. The negative terminal of the array was connected to the collector of two NPN N2222 transistors in parallel. The signal pins of each transistor were then connected to pin 9 of the microcontroller. The UNO was programmed to send a signal through pin 9 to the transistors, activating the light box for a set duration at specific time intervals (e.g., turn on for 1 min once every hour). PWM pin 9 was used so that one could turn on the array at varying intensities. At maximum intensity, the light array achieved an illumination of 1 mW/m² as measured by a photodiode power sensor (S120C, Thorlabs) placed at the sample distance.

In order to adjust the timing and intensity of light exposure, an HC-06 ARDUINO® Bluetooth module was installed, allowing the device to communicate with an Android™ smartphone application made with MIT App Inventor. The application allows the user to set the length of exposure, time between exposures, number of exposures, the delay time before the exposure cycles began, and the relative intensity of the LED array.

Plates containing the samples that were to be illuminated were placed onto the light array directly after transfection in the 37 °C humidified incubator and a sheet of aluminum foil was used to cover the entire apparatus. The plates not treated for light illumination were wrapped in aluminum foil directly after transfection. These dark plates were either kept constantly in the 37 °C incubator until flow cytometry

or removed starting at 18 h post transfection and placed outside the incubator at room temperature (22 °C) or in a refrigerator at 4 °C; these temperature pulses were performed four times, once every hour for 15 min durations.

**Enzyme-linked immunoabsorbance assay**. Cytokine concentrations for Fig. 5b were measured using enzyme-linked immunoabsorbance assays from Becton Dickinson (cat. no. 555142, 555190, 555157, and 550534). Absorbance measurements of supernatant cytokine concentration were made using a Molecular Devices SpectraMax M5 plate reader at 450 nm with wavelength correction at 570 nm.

**Flow cytometry**. For all cytometry experiments, except for given times during time-course experiments, plates were removed from the incubator 2 days post transfection. Cells were trypsinized using 0.05% trypsin/0.53 mM EDTA (Corning) and resuspended with cell culture medium to a volume of 200 μL in 96 U-bottom plates. All cell populations were analyzed using a Thermo Fisher Attune Nxt cytometer equipped with four lasers. GFP was detected using a 488 nm blue laser and a 510/10 bandpass emission filter; a 530/30 bandpass filter was used for experiments that converted to MEFL units. mTagBFP was detected using a 405 nm violet laser and 440/50 bandpass emission filter. iRFP720 was detected using a 638 nm red laser and 720/30 bandpass emission filter. Experiments in Fig. 2 were analyzed using TASBE Flow Analytics, an open source flow cytometry software tool[27,35]. In short, fluorescence channel measurements were converted to units of MEFL using 8-peak rainbow beads (Spherotech RCP-30-5A) and using linear color translation and compensation between fluorophores. Gaussian mixture models were used to isolate viable cell populations and remove debris. Results were analyzed above a cutoff of $10^6$ MEFL on the constitutive transfection marker. For all other experiments, results were analyzed using FlowJo 9 (TreeStar), using a polygonal FSC-A/SSC-A gate to remove debris from the mammalian cell populations. An mTagBFP fluorescent protein was used as a transfection marker and positive BFP cells were first gated, and then mean GFP fluorescence was determined.

**Xenograft mouse models**. Female NSG mice, 4–6 weeks of age, were purchased from Jackson Laboratories (#005557) and maintained in the BUMC Animal Science Center. All xenograft mouse protocols comply with ethical regulations for animal research and were approved by the Institutional Animal Care and Use Committee at BUMC (18-005). In order to carry out the xenograft models, NSG mice were initially injected with $20 \times 10^6$ of transiently transfected HEK293FT cells subcutaneously. After 24 hr, mice were randomly grouped, and different drugs were injected intraperitoneally (200 mg/kg ABA, 20 mg/kg GIB ester, 0.06 mg/kg rapalog). After 24 h of drug injection, luciferase signal was measured by IVIS Spectrum (Xenogen) and was quantified as total flux (photons per s) in the region of interest to verify split-recombinase activity. Images were acquired within 30 min following intraperitoneal injection of 150 mg/kg of D-luciferin (Perkin-Elmer, #122799).

**Statistical methods**. All transient gene expression experiments involved transfection of DNA into $n = 3$ separate cell cultures. Fluorescence intensities for each cell culture population was averaged and the standard error of mean or standard deviation was taken as noted.

**Reporting summary**. Further information on research design is available in the Nature Research Reporting Summary linked to this article.

## Data availability
Data are available from the corresponding authors upon reasonable request. Source data are available in the Source Data file. Plasmids and construction resources are available at https://www.addgene.org/browse/article/28193023

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

## Acknowledgements

B.H.W. acknowledges funding from the NSF Graduate Research Fellowship Program (DGE-1247312), a NIH/NIGMS fellowship (T32-GM008764) and the NSF Expeditions in Computing Award No. 1522074, which is part of the Living Computing Project (https://www.programmingbiology.org/). W.W.W. acknowledges funding from the NIH Director's New Innovator Award (1DP2CA186574), NSF Expeditions in Computing (1522074), NSF CAREER (162457), NSF BBSRC (1614642), NSF EAGER (1645169), and Boston University College of Engineering Dean's Catalyst Award. J.B. acknowledges funding from the NSF Expeditions in Computing (1522074) grant. We thank D. Densmore and D. Chakravarti for suggestions on the manuscript. We acknowledge S. Koplik and T. Vantrimpont for contributions to preparations of molecular reagents, and Fu-Sen Liang and Hao Yan from the University of New Mexico for their generous help with providing some ABA-related reagents.

## Author contributions

Conceptualization, B.H.W. and W.W.W.; methodology, B.H.W. and B.L.; software, W.K. J.B. and J.B.; validation, B.H.W., C.J.H., Y.A. and N.T.; Formal Analysis, B.H.W., J.H.C., Y.A., J.L., and J.B.; investigation, B.H.W., J.H.C., Y.A., N.T., M.W., M.T., C.O., N.T. and B.L.; resources, B.H.W., Y.A., N.T.H.P., L.D.C., N.T., M.W., M.T., C.O., B.L., J.L., and W.K.J.B.; writing—original draft, B.H.W.; writing—review and editing, B.H.W., J.H.C., N.T.H.P., J.B., and W.W.W.; visualization, B.H.W.; Supervision, B.H.W., J.B., and W.W.W.; funding acquisition, B.H.W., J.B., and W.W.W.

## Competing interests

The authors declare the following competing interests: Boston University has filed a patent application, USPTO 15/838,598, (Inducible dimerization of recombinases) with B. H.W. and W.W.W. as the named inventor based on this work. W.W.W. has consulted for and own shares in Senti Biosciences. B.H.W. and W.W.W. have no other competing interests. The other authors declare no competing interests.
