## [Peer Review File · Nature Communications]

Reviewers' Comments:

Reviewer #1:

Remarks to the Author:

This revised manuscript is an improved version. The authors have done well to apply their tools in reasonable application settings. Overall this reviewer feels the manuscript has fairly addressed previously raised concerns.

Reviewer #2:

Remarks to the Author:

Weinberg et al. added an in vivo application shown in Figure 5 for manuscript revision. The reviewer recognizes this application as significant data. However, in vivo applications of inducible DNA recombinases other than split-Cre, that is, Flp, have recently been published by Heo group (Nat. Commun. (2019) 20: 34), which appears to affect the revised manuscript. Unfortunately, the authors did not refer this publication in the revised manuscript.

“High-performance chemical and light-inducible recombinases in mammalian cells and mice”
(NCOMMS-19-26083-T)

REVIEWERS' COMMENTS:

Reviewer #1 (Remarks to the Author):

This revised manuscript is an improved version. The authors have done well to apply their tools in reasonable application settings. Overall this reviewer feels the manuscript has fairly addressed previously raised concerns.

Thank you for the comments.

Reviewer #2 (Remarks to the Author):

Weinberg et al. added an in vivo application shown in Figure 5 for manuscript revision. The reviewer recognizes this application as significant data. However, in vivo applications of inducible DNA recombinases other than split-Cre, that is, Flp, have recently been published by Heo group (Nat. Commun. (2019) 20: 34), which appears to affect the revised manuscript. Unfortunately, the authors did refer this publication in the revised manuscript.

During our revision, Heo and colleagues published a very nice work on using a light-inducible Flp in the mouse brain, which we will reference in our manuscript (ref 4 in the introduction) along with some of the existing inducible recombinases. This work highlights the importance of having even 1 new inducible recombinase on mammalian research. Our collection of >20 inducible recombinases and a curated set of split sites will enable further development of advanced animal studies.